# An Innovative Topical Medical Device with Hyaluronic Acid and Polypeptides in Patients with Reduced Knee Function

**DOI:** 10.3390/jfmk9010031

**Published:** 2024-02-15

**Authors:** Tommaso Bonanzinga, Alice Giulia De Sensi, Beatrice Balzarini, Gian Luca Doro, Luca Bertolino, Luca Forte, Elizaveta Kon

**Affiliations:** 1Department of Biomedical Sciences, Humanitas University, Pieve Emanuele, 20072 Milan, Italy; tommaso.bonanzinga@hunimed.eu (T.B.); beatrice.balzarini@humanitas.it (B.B.); gianluca.doro@humanitas.it (G.L.D.); luca.bertolino@st.hunimed.eu (L.B.); elizaveta.kon@humanitas.it (E.K.); 2IRCCS Humanitas Research Hospital, Rozzano, 20089 Milan, Italy; 3Contrad Swiss SA, 6900 Lugano, Switzerland; luca.forte@contrad.ch

**Keywords:** osteoarthritis, hyaluronic acid, knee, topical, musculoskeletal, HA

## Abstract

A topical medical device, AI500^®^, constituted of a single-chain polypeptide embedded in hyaluronic acid, was tested and evaluated in patients with reduced knee function due to osteoarthritis and other knee conditions. A total of 35 participants with reduced knee function assessed by the WOMAC Physical Function score were recruited. Four study visits were planned, from the first application at V0 to 1 week follow up at V3. Patient symptomatology was evaluated after 24 h (V1) and after 48 h (V2) through phone contact, and after 1 week from V0, on site (V3). The overall duration of the follow up was one week. An amelioration of 40% in WOMAC Physical Functional scores after 1 week of treatment was recorded, thus achieving the primary endpoint of 20%. Furthermore, a reduction of 29% in Physical Functional scores and of 28% in total WOMAC scores between V0-V2 was registered, together with a decrease of 39% between V0 and V3. The NRS scale showed a 29% and 37% reduction in pain between V0-V1 and V0-V2, respectively. Product safety was confirmed by the very low rate of adverse effects, non-device related, observed in only 2 patients out of 35, resolved spontaneously within 24–48 h. No safety concerns or risks associated with the use of the device were highlighted. There are few the studies on the topical use of HA-based gels for the treatment of knee problems. Compared to invasive intra-articular injections and oral pharmacological therapies used in cases of knee pain, the topical application of AI500^®^ is non-invasive, safe, and appreciated by patients. Good results in terms of functional improvement and symptoms resolution were obtained in less than 1 week.

## 1. Introduction

Knee pain is a very common condition encountered in clinical practice, affecting up to 25% of the adult population [1]. Due to the increasing incidence of its presentation, it has become an extremely frequent and disabling chronic condition, especially in Western populations [2]. The most common causes of knee pain include ligamentous lesions—e.g., collateral ligaments and anterior cruciate ligaments sprain or rupture [3,4]—meniscal tears [5], patello-femoral syndrome [6], ilio-tibial band syndrome and other tendinopathies, e.g., pes anserinus tendinitis [3], and osteoarthritis [7]. The resulting low quality of life, functional limitations and severe pain push individuals to seek medical care [8].

As most of these causes could be radically solved only with surgical treatment [9], the goal of the multiple conservative strategies nowadays available is—in addition to offering relief of symptoms, locally reduce inflammatory distress and improve joint function—to delay surgery [10].

Pharmacological therapies generally include non-steroidal anti-inflammatory (NSAIDs) products such as ibuprofen or new generation COX-inhibitors [11] as well as analgesics such as paracetamol. Especially in chronic pain, prolonged intake of drugs can be responsible for a number of non-negligible side effects [12]. For this reason, topically acting drugs have been developed and shown to be effective for the treatment of musculoskeletal pain and, albeit to a more limited extent, in chronic pain [13,14]. Similarly, topical treatments can cause side effects ranging from local irritation to undesired systemic absorption of the active ingredient [15,16]. To remedy these possible consequences, the use of hyaluronic acid (HA)—especially in the form of intra-articular injection—has been introduced, as it is naturally produced by the body and diffused in all tissues. It is a non-sulfate glycosaminoglycan present in large quantities in cartilage extracellular matrix and synovial fluid (SF), providing it with viscoelastic and lubricating characteristics in slow joint movements [17] and shock-absorbing characteristics in fast joint movements [18]. Moreover, as it is also a biocompatible, biodegradable, non-immunogenic and non-inflammatory compound, it has been widely used in the treatment of musculoskeletal pathologies.

Although the validity and good results of HA intra-articular infiltrations are now well known, their first use date back almost 30 years [19] and they bring with them some risks that should not be underestimated, such as acute local reactions, infection and residual pain [20,21]. In order to avoid the potential side effects associated with injections, HA can be used as a topical treatment, for it exploits a useful non-invasive approach. As a matter of fact, when combined with other peptide adjuvant factors, such as cytokines or growth factors, the cutaneous absorption of HA is greatly enhanced [22], ensuring the penetration of the product into the joint and the local completion of its physiological lubricating functions [19]. Among these peptides, SH-Polypeptide-6 plays a key role: it is a synthetic single-chain peptide synthesized starting from a human gene coding for interleukin-10 (IL-10), the latter known for its anti-inflammatory effects on several immune cell types, limiting excessive tissue degeneration caused by inflammation [23]. In addition, IL-10 demonstrated beneficial effects on pain modulation too. In actual fact, it generates anti-nociceptive effects through the expression of microglial beta-endorphins, thus reducing neuropathic pain via the up-regulation of IL-27 [24]. The combination of the molecular weight of the HA (1 × 10^6^ and 1.8 × 10^6^ Daltons) with the peptides represent the integral part of the technology that facilitates its delivery through the skin, thus accounting for an innovative in situ treatment intended to provide relief in cases of muscular pain and mobility limitations [22,25,26].

The results reported in a recent paper by Masotti et al. [25] showed that topical hyaluronic acid and biomimetic recombinant peptide formulations can positively contribute to the treatment of myalgia and joint pain. Based on such evidence, in this study we evaluated the efficacy of the topical medical device, AI500^®^, to promote functional improvement and pain resolution in acute new-onset knee conditions and chronic knee pain flare-ups within 1 week of treatment.

## 2. Materials and Methods

The study is a monocentric, post-market confirmatory interventional, single arm clinical investigation conducted at Humanitas Research Hospital, Milan, Italy, between 11 July 2022 and 10 May 2023, approval code NCT05886608. Once informed about the risks and benefits provided by the study, enrolled subjects freely signed an Informed Consent (IC) approved by the Ethical Committee, according to the principles of the Declaration of Helsinki. Consent to the use of patients’ data was in line with European laws regarding the use of personal data (EU 2016/679) and local Italian laws (196/2003). A total of 35 participants with a mean age of 55.5 years ± 15.9 years with reduced knee function on WOMAC Physical Function score were recruited for the study. Patients were screened for eligibility based on the inclusion/exclusion criteria reported in Table 1. Final demographic characteristics of the patients included in the study is reported in Table 2.

The evaluated product (AI500^®^, Contrad Swiss SA, Switzerland) is a medical device, based on water and glycerin, single-dose gel, containing sodium hyaluronate and a peptide mixture inside a 1.5 mL single-dose vial. The gel was applied on the skin directly to the painful site with a patch in order to guarantee the product absorption. A single batch n.20A04 was used.

For each subject, four study visits were planned: During the baseline visit (V0) the enrolled subjects had to sign the ICF and were treated with the first application of AI500^®^ performed by the Clinician. The following day, the second application was performed by patients themselves at home. Patients’ symptomatology was evaluated after 24 h (V1) and after 48 h (V2) through phone contact, and after 1 week from V0, on site (V3). The overall duration of the follow up was 1 week. Potential adverse events (AEs), concomitant medications and device deficiencies/incidents were monitored at each visit. The primary endpoint of our study was to evaluate the clinical performance of AI500^®^ in improving knee function after 1 week of treatment and to assess the changes from baseline (V0) to week 1 (V3) in the WOMAC Physical Function score. The secondary endpoints were also to evaluate WOMAC Physical Function at 48 h, total WOMAC score between 48 h and 1 week from treatment, pain intensity on a Numerical Rating Scale (NRS) at 24 and 48 h, safety (adverse events) and patient satisfaction (5-point Likert scale).

Considering a minimum difference of 20% in terms of WOMAC Physical Function (equal to 10 points) between week 1 and the baseline visit—with a standard deviation (SD) equal to 15, a medium Pearson correlation of 30% between two timepoints and a type I error of 5%—31 patients were sufficient to reach a statistical power greater than 80%. Planning to enroll a total of 35 patients would allow for a 10% drop-out rate. Statistical power calculation was performed using SAS^®^ software, version 9.4. To analyze the primary endpoint, descriptive statistics were presented and relative differences were calculated between baseline and 1-week-after WOMAC Physical Functional Score. The Kolmogorov–Smirnov test was used to check the normality of distribution of the WOMAC Physical Function sub-score at each visit: in case the *p*-values were >0.05, the distribution was considered normal and a paired t-test was applied to verify the difference between baselines and final values; when *p*-values were <0.05, a Wilcoxon test for paired data was applied to test the difference between baselines and final values. Statistical analyses were performed using Graphpad Prism^®^ 10.1.2.

Secondary outcomes were reported descriptively at each study visit as appropriate. The evaluation of adverse events and physical examination were performed using SAS^®^ software, version 9.4.

## 3. Results

At V0 and V3 a knee examination—including inspection, palpation, range of motion, strength, stability, neuromuscular assessment and special provocative tests—was performed in all (100%) patients included in the study.

### 3.1. Primary Efficacy Analysis

After one week of treatment, a decrease of 40% (*p* < 0.0001) in the WOMAC Physical Functional scale score between V0 and V3 was registered, as the mean value went from 26.9 ± 6.4 at V0 to 16.1 ± 10.8 at V3. The expected endpoint of 20% improvement was, therefore, achieved (Table 3, Figure 1).

### 3.2. Secondary Efficacy Analysis

#### 3.2.1. Knee Function Improvement from V0 to V2

Table 3 shows the WOMAC physical function sub-scores registered at V0, V2 and V3 and the difference between V0-V2 and V0-V3. In addition to V0-V3, a statistically significant improvement by 29% was also demonstrated between V0 and V2 (*p* < 0.0001). Moreover, a significant further reduction between V2 and V3 (*p* = 0.0113) was recorded.

#### 3.2.2. WOMAC Total Score

At V2 and V3, a decrease of 28% and 39% was shown, respectively, in terms of the WOMAC total index score with respect to V0 (*p* < 0.0001). The decrease from V2 to V3 in total WOMAC score was also statistically significant (*p* = 0.0111) (Table 4, Figure 2).

#### 3.2.3. Pain Alleviation after 24 h and 48 h of Treatment

Changes in NRS between V0-V1 and V0-V2 were analyzed (Table 4). Mean NRS scores went from 5.5 ± 1.4 at V0 to 3.9 ± 1.4 at V1 and 3.5 ± 1.5 at V2, corresponding to a reduction of 29% and 37% between V0-V1 and V0-V2, respectively (*p* < 0.0001), with a statistically significant difference also between V1 and V2 (*p* = 0.0144) (Table 5, Figure 3).

#### 3.2.4. Patient Satisfaction

A 5-point Likert Scale was used at the End of Study visit (V3). Satisfaction, measured by a 5-point Likert scale, was calculated based on the feedback of the 35 patients: the mean value was 3.7 ± 0.9 (median value was 4.0), with a minimum value of 2.0 and a maximum value of 5.0 points.

### 3.3. Adverse Events and Adverse Devices Effects

During the study, only two patients had at least one Adverse Event (AE), but none of the patients had a device-related AE; moreover, serious or severe AEs were absent. In particular, one patient reported acute knee pain and knee stiffness, which was resolved without complications in one day. Nagging pain behind the knee, mild knee pain and mild knee heaviness were also reported by another candidate; also, in this case AEs were resolved within 2 days. All AEs were classified as “mild” and not serious; consequently, no AE led to study termination or death. A detailed report of AE is listed in Table 6.

## 4. Discussion

In the present study, a topical medical device—AI500^®^—composed of hyaluronic acid and SH-Polypeptide-6 was evaluated. Validity in terms of functional limitations improvement and pain resolution was considered and tested. Patients reporting knee pain and functional limitations were enrolled in the study and were visited at Humanitas Research Hospital, Rozzano (MI), by the same Clinician over a time period of 11 months.

Although topical NSAIDs such as Diclofenac or Ketoprofen showed good results in terms of pain alleviation [27]—yet carrying with them limited efficacy in certain cases of KOA [14]—other HA-based devices have previously proved to be useful for the treatment of pain associated with KOA and for knee functional improvement [28]. Similarly to the latter, after data collection and statistical analysis, the overall study results were fully satisfying for all the considered knee pain conditions treated with AI500^®^. For what concerns performance of the IP in terms of knee function after 1 week of treatment, the study showed a decrease of 40% in WOMAC Physical Functional scores between V0 and V3, thus achieving the primary target previously set at 20% at the beginning of the study. Knee function improvement after 48 h of treatment (V0-V2) showed a statistically significant amelioration, as well as between V2 and V3. Variations in pain intensity through the NRS evaluated at baseline (V0), 24 h after baseline (V1) and 48 h after baseline (V2) showed a notable improvement.

Regarding overall patients’ satisfaction, AI500^®^ showed remarkable results for all patients participating to the study.

Ultimately, the safety of the product was evaluated as well and it showed a very low percentage of AEs, not device related, all of them resolved spontaneously within 24–48 h. Moreover, no safety concerns or risks associated with the use of the device were highlighted.

Based on the above evidence, we validated the efficacy of AI500^®^ in enhancing knee functionality, with improvements already observed one week post-treatment. As NRS pain intensity significantly improved after 48 h of treatment, patients were overall satisfied with the studied product. Based on these results, we can support the benefits of using AI500^®^ in patients with reduced knee function.

## 5. Conclusions

Currently, there are few published studies on the topical use of HA-based gels for the treatment of knee problems. This report shows the results of a clinical investigation on AI500^®^ in patients with reduced knee function and highlights the optimal outcomes obtained in terms of knee functionality improvement, pain resolution and product safety.

Compared to invasive knee injections of HA, the topical application of AI500^®^ is non-invasive, safe, convenient to use and appreciated by patients. Furthermore, topical application of the medical device can require less than 1 week to see improvements in knee function and pain.

This study has some limitations among which the low number of patients and the absence of a control group. Future studies will investigate whether the positive effects of AI500^®^ persist beyond the treatment period.

## Figures and Tables

**Figure 1 jfmk-09-00031-f001:**
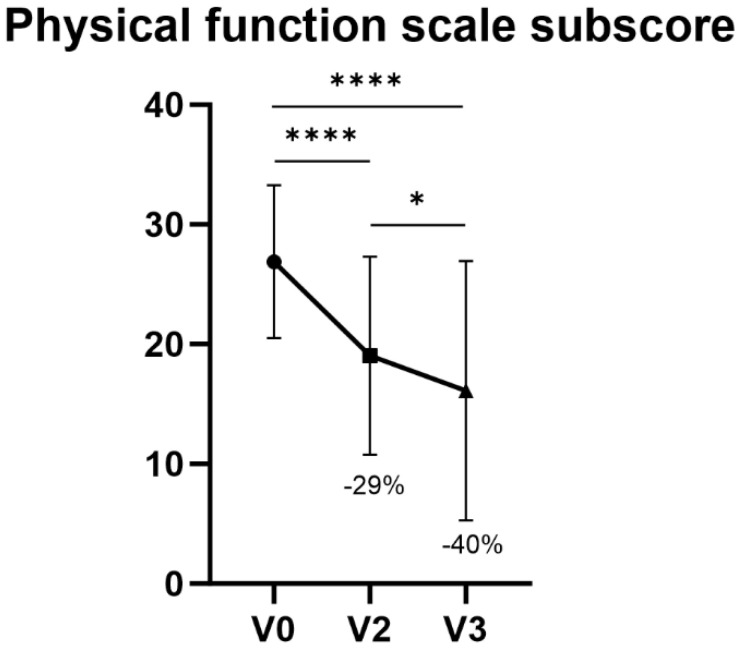
Physical function scale subscore. **** *p* < 0.001, * *p* < 0.05.

**Figure 2 jfmk-09-00031-f002:**
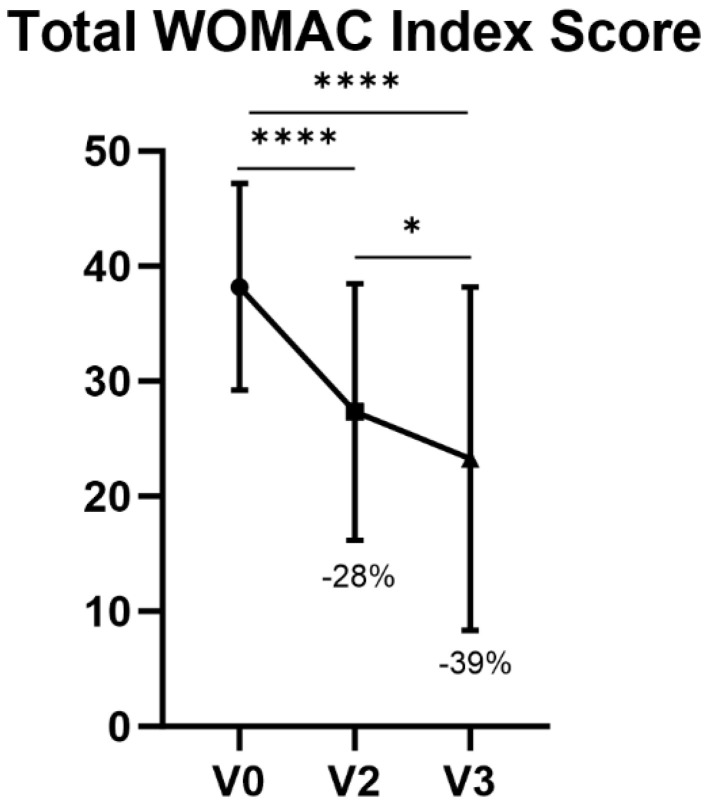
Total WOMAC Index Score. **** *p* < 0.001, * *p* < 0.05.

**Figure 3 jfmk-09-00031-f003:**
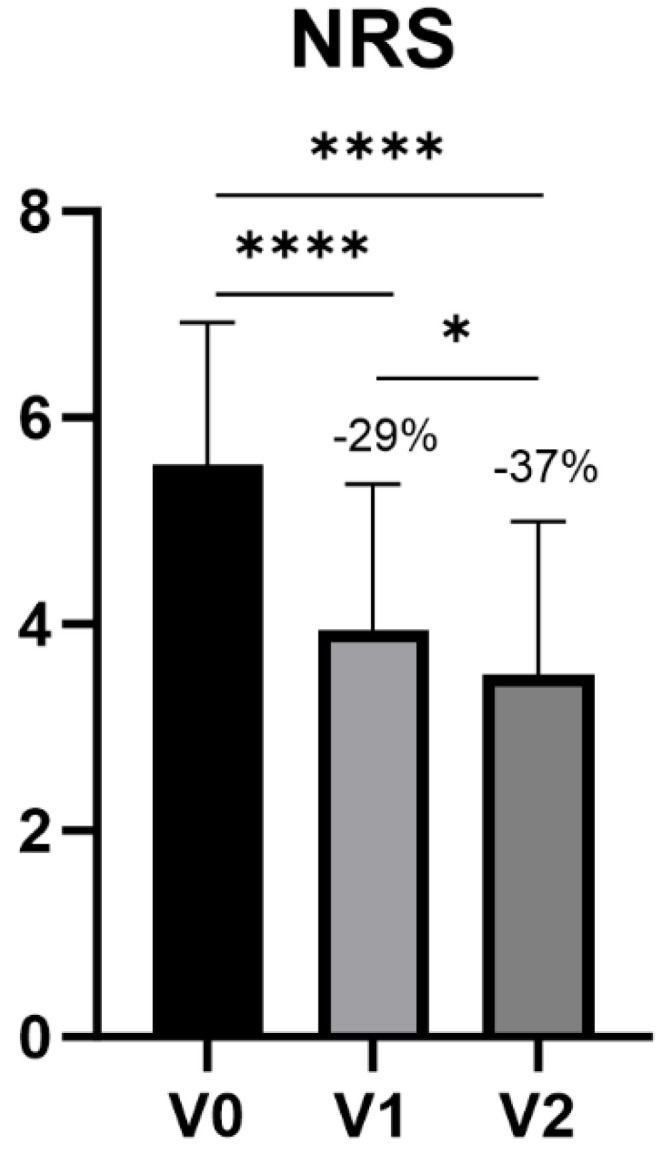
NRS. **** *p* < 0.001, * *p* < 0.05.

**Table 1 jfmk-09-00031-t001:** Inclusion and exclusion criteria.

Inclusion Criteria	1. Signed patient informed consent form (ICF). 2. Male or Female aged ≥ 18 years at the time of the signature of ICF. 3. Presenting with reduced knee function caused by osteoarthritis flare-ups, meniscal injuries, ligament injuries, inflammation of soft tissues, assessed as 20–45 rating according to the WOMAC function scale. 4. Willingness to follow all study procedures, including attending all site visits, tests and examinations. 5. Willingness to follow indications.
Exclusion Criteria	Use of analgesics within the 24 h prior to V0. Damaged, infected or ulcerated skin in the area of treatment. Ongoing cutaneous allergies. Serious and chronic pathological skin conditions (i.e., rosacea, psoriasis, vitiligo) or lesions including cancer with/without ongoing antitumor therapy. Patients suffering from muscular dystrophy. Patients presenting with bone fractures or severe injuries (including locked knee). Severely disabled arthritic patients using a wheelchair. Allergy to device components (Sodium hyaluronate; SH-Polypeptide-6; Glycerin; Propylene glycol; Ethylhexylglycerin; Panthenol; PEG-40 hydrogenated castor oil; Sodium hydroxide; Xanthan gum; Phenoxyethanol; Benzoic Acid; Carbomer; Dehydroacetic Acid; Disodium EDTA). Immune system illnesses. Uncontrolled systemic diseases. Known drug and/or alcohol abuse. Mental incapacity that precludes adequate understanding or cooperation. Participation in another investigational study. Pregnancy or breastfeeding. Patient with both knees affected/damaged.

**Table 2 jfmk-09-00031-t002:** Patient characteristics.

		Overall (N = 35)
Age (years)	Mean ± SD	55.5 ± 15.9
	Median (Min–Max)	60.0 (21.0–81.0)
	Missing	0
Gender		
Male	N (%)	12 (34.3)
Female	N (%)	23 (65.7)
Missing	N	0
Ethnic group		
Caucasian	N (%)	34 (97.1)
Other	N (%)	1 (2.9)
Smoking status		
No smoker	N (%)	31 (88.6)
Former smoker	N (%)	2 (5.7)
Smoker	N (%)	1 (2.9)
Smoker with medical device	N (%)	1 (2.9)
Missing	N	0
Alcohol consumption		
No	N (%)	32 (91.4)
Yes	N (%)	3 (8.6)
Missing	N	0
Pregnancy test results		
Negative	N (%)	5 (14.3)
NA/Male	N (%)	30 (85.7)

**Table 3 jfmk-09-00031-t003:** WOMAC physical function score.

WOMAC Physical Function Subscore	N	Mean	(SD)	Med	(IQR)	Min	−Max	P^1^
V0	35	26.9	(6.4)	25.0	(7.0)	20.0	−44.0	
V2	35	19.1	(8.3)	18.0	(7.0)	−3.0	−43.0	<0.0001
Difference between V0 and V2	35	−7.8	(3.9)	−8.0	(4.0)	−20.0	−1.0	
V3	35	16.1	(10.8)	13.0	(12.0)	3.0	−44.0	
Difference between V0 and V3	35	−10.8	(6.5)	−12.0	(11.0)	−21.0	−1.0	<0.0001

**Table 4 jfmk-09-00031-t004:** WOMAC total score.

WOMAC Total Score	N	Mean	(SD)	Med	(IQR)	Min	−Max	P^1^
V0	35	38.2	(9.0)	36.0	(11.0)	27.0	−62.0	
V2	35	27.3	(11.3)	28.0	(11.0)	8.0	−58.0	<0.0001
Difference between V0 and V2	35	−10.9	(5.1)	−10.0	(7.0)	−22.0	−2.0	
V3	35	23.3	(14.9)	17.0	(19.0)	6.0	−62.0	
Difference between V0 and V3	35	−15.0	(8.5)	−17.0	(13.0)	−27.0	−1.0	<0.0001

**Table 5 jfmk-09-00031-t005:** Numerical Rating Scale (NRS).

NRS Score	N	Mean	(SD)	Med	(IQR)	Min	−Max	P^1^
Visit	35	5.5	(1.4)	5.0	(2.0)	2.0	−8.0	
V0	35	27.3	(11.3)	28.0	(11.0)	8.0	−58.0	
Difference between V0 and V1	35	−1.6	(1.4)	−2.0	(2.0)	−6.0	−1.0	<0.0001
V1	35	3.9	(1.4)	4.0	(2.0)	2.0	−6.0	
Difference between V0 and V2	35	−2.0	(1.4)	−2.0	(2.0)	−6.0	−1.0	<0.0001
V2	35	3.5	(1.5)	3.0	(3.0)	1.0	−7.0	

**Table 6 jfmk-09-00031-t006:** Number of Adverse Events (AE).

Number of Patients with at Least One AE	N (%)	2 (5.7)
Number of patients with at least one device-related AE	N (%)	0 (0.0)
Number of patients with at least one serious AE	N (%)	0 (0.0)
Number of patients with at least one severe AE	N (%)	0 (0.0)
Number of patients who prematurely terminated the study due to an AE	N (%)	0 (0.0)
Number of deaths	N (%)	0 (0.0)
Number of AEs	N	6
Number of SAEs	N	0
Number of related AEs	N	0
Number of severe AEs	N	0
Number of deaths	N	0
		Overall (N = 6)
Arthralgia	N (%)	5 (83.3)
Joint stiffness	N (%)	1 (16.7)

## Data Availability

In order to guarantee privacy protection, the raw data supporting the conclusions of this article will be made available by the authors on request.

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
