# Peer review of "An Innovative Topical Medical Device with Hyaluronic Acid and Polypeptides in Patients with Reduced Knee Function"

_jfmk, 2024, doi:10.3390/jfmk9010031_

Round 1

Reviewer 1 Report

Comments and Suggestions for Authors

The results correspond to the clinical trial no. NCT05886608. Thus, I do believe this article should be published.

In the introduction it could be excellent if the authors affirm/confirm the role of the molecular weight of the polysaccharide. Also, the identification of the batches is missing.

In this idea, several batches of the product were used, or all the patients were treated with the same. Please clarify.

Author Response

Dearest, I would like to thank you sincerely for taking the time to read and correct our manuscript. Your kind feedback is very important to us as it is crucial for the collective growth of our scientific community.

Regarding the two comments you submitted to us, you can find our answers listed here: 

- A single batch of product was used during the clinical study, specifically the batch was n.20A04

- The molecular weight of hyaluronic acid is between 1x10^6 and 1.8x10^6 Daltons. As for the role of MW, the combination of that type of HA with the peptides is the integral part of the technology that facilitates its delivery through the skin. 

I added both details in the manuscript, as you suggested.

I sincerely hope that I have been able to answer your questions completely. Please do not hesitate to point out further clarifications in case you have further doubts.

Thank you again for your kind help.

Reviewer 2 Report

Comments and Suggestions for Authors

An article describes a topical medical device  constituted by a single-chain polypeptide embedded in hyaluronic acid, tested  in 35 patients with  knee osteoarthritis . A topical and effective route of drug administration can be an alternative to the intraarticular or oral ways. The method of drug administration proposed by the authors should be further investigated. In my opinion an article:"An innovative topical medical device with hyaluronic acid and polypeptides in patients with reduced knee function" should by publish in Journal of FAMS.

Author Response

Dearest, I would like to thank you sincerely for taking the time to read and correct our manuscript. Your kind feedback is very important to us as it is crucial for the collective growth of our scientific community.